# Ultrasound-Guided Interventions for Carpal Tunnel Syndrome: A Systematic Review and Meta-Analyses

**DOI:** 10.3390/diagnostics13061138

**Published:** 2023-03-16

**Authors:** King Hei Stanley Lam, Yung-Tsan Wu, Kenneth Dean Reeves, Felice Galluccio, Abdallah El-Sayed Allam, Philip W. H. Peng

**Affiliations:** 1The Department of Clinical Research, The Hong Kong Institute of Musculoskeletal Medicine, Hong Kong; 2Faculty of Medicine, The Chinese University of Hong Kong, Hong Kong; 3Faculty of Medicine, The University of Hong Kong, Hong Kong; 4Center for Regional Anesthesia and Pain Medicine, Wan Fang Hospital, Taipei Medical University, Taipei 110, Taiwan; 5Center for Regional Anesthesia and Pain Medicine, Chung Shan Medical University Hospital, Taichung 402, Taiwan; 6Department of Physical Medicine and Rehabilitation, Tri-Service General Hospital, School of Medicine, National Defense Medical Center, Taipei 114, Taiwan; 7Integrated Pain Management Center, Tri-Service General Hospital, School of Medicine, National Defense Medical Center, Taipei 114, Taiwan; 8Department of Research and Development, School of Medicine, National Defense Medical Center, Taipei 114, Taiwan; 9Private Practice PM&R and Pain Management, 4840 El Monte, Roeland Park, KS 66205, USA; 10Fisiotech Lab. Studio, Rheumatology and Pain Management, 50136 Firenze, Italy; 11Morphological Madrid Research Center (MoMaRC), 10107 Madrid, Spain; 12Department of Physical Medicine, Rheumatology and Rehabilitation, Faculty of Medicine, Tanta University, Tanta 31527, Egypt; 13Clinical Neurophysiology Fellowship, Arab Board of Health Specializations, Ministry of Health, Baghdad 61298, Iraq; 14Department of Anesthesiology and Pain Medicine, The University of Toronto, Toronto, ON M5T 2S8, Canada

**Keywords:** carpal tunnel syndrome, ultrasound-guided, intervention, injection, release

## Abstract

Carpal tunnel syndrome (CTS) is the most common peripheral entrapment, and recently, ultrasound-guided perineural injection (UPIT) and percutaneous flexor retinaculum release (UPCTR) have been utilized to treat CTS. However, no systematic review or meta-analysis has included both intervention types of ultrasound-guided interventions for CTS. Therefore, we performed this review using four databases (i.e., PubMed, EMBASE, Scopus, and Cochrane) to evaluate the quality of evidence, effectiveness, and safety of the published studies on ultrasound-guided interventions in CTS. Among sixty studies selected for systemic review, 20 randomized treatment comparison or controlled studies were included in six meta-analyses. Steroid UPIT with ultrasound guidance outperformed that with landmark guidance. UPIT with higher-dose steroids outperformed that with lower-dose steroids. UPIT with 5% dextrose in water (D5W) outperformed control injection and hydrodissection with high-volume D5W was superior to that with low-volume D5W. UPIT with platelet-rich plasma outperformed various control treatments. UPCTR outperformed open surgery in terms of symptom improvement but not functional improvement. No serious adverse events were reported in the studies reviewed. The findings suggest that both UPIT and UPCTR may provide clinically important benefits and appear safe. Further treatment comparison studies are required to determine comparative therapeutic efficacy.

## 1. Introduction

Carpal tunnel syndrome (CTS) is the most commonly diagnosed (90%) peripheral entrapment neuropathy [1]. The worldwide incidence of CTS ranges from 3% to 4%, and its typical symptoms and signs include numbness, tingling, pain or burning sensation, and nocturnal paresthesia in the regions innervated by the median nerve (MN). Weakness with thenar muscle atrophy may occur in severe cases or in later stages in mild-to-moderate cases [2,3]. Although the pathophysiology of CTS remains unclear, it is generally acknowledged to involve MN compression by increased pressure within the carpal tunnel caused by swollen flexor retinaculum (FR), flexor tenosynovium, or subsynovial connective tissue (SSCT), or a combination thereof. Increased intracarpal pressure contributes to interruption of nerve microcirculation, ischemia, impaired nerve conduction, decreased MN dynamics with adhesion, increased MN vascular permeability, and interruption of the axoplasmic flow, with subsequent nerve swelling proximal and distal to the compression site [4,5,6]. The “hourglass” configuration distortion of the MN could further decrease the MN excursion, thereby exacerbating traction neuropathy [7].

The treatment of CTS includes non-surgical and surgical management depending on symptom severity [4,8]. Generally, non-surgical treatments (such as splints, medications with nonsteroidal anti-inflammatory drugs, and physical therapy) are recommended for mild-to-moderate CTS. However, non-surgical treatments have a limited therapeutic effect with a short duration [9]. A recent systemic review revealed that 57–66% of patients underwent surgery after receiving non-surgical treatments for 1–3 years [1]. An excellent outcome after CTS surgery was reported in 75% of patients, with symptom escalation in 8% of patients, and revision surgeries in 12% of patients [10]. However, revision surgery is associated with a dramatic increase in the failure rate [10,11,12]. The percentage of unfavorable surgical outcomes primarily due to an incomplete release of the FR or scar formation with resultant grasp weakness or pillar pain has been reported to be consistent since 1988 [12]. Conventionally, carpal tunnel release (CTR) is performed using open CTR (OCTR), endoscopic CTR (ECTR), or mini-open CTR (mini-OCTR) approaches. Overall, OCTR and ECTR have similar percentages of successful surgical outcomes and associated complications [13,14]. OCTR has the advantage of enabling clear visualization of the essential anatomic structures; however, it typically requires a 2–7 cm palmar skin incision [15]. The time to return to work for OCTR is typically 3 to 4 weeks [13,16]. ECTR requires one or two portals of entry, each via a smaller (1.5–2 cm) incision than that required by OCTR, leading to fewer scar-related complications and a shorter duration of work loss (2–3 weeks). However, ECTR is associated with a higher rate of transient nerve injury [15]. Moreover, mini-OCTR also aims to minimize surgical trauma and has shown positive results, but a portion of the surgery is performed blindly [17,18].

High-resolution ultrasound has been increasingly used to administer nerve blocks because the direct visualization of nerves enables more precise, effective, and safe anesthetic infiltration while limiting the potential for neural trauma [19,20]. Recently, ultrasound-guided perineural injection and percutaneous FR release have been utilized to treat CTS, with reported clinically and statistically significant benefits [21,22]. Although a review in 2022 mentioned several potential injectates for use in the perineural injection therapy (PIT) of CTS [23], the only systematic collection of outcomes of all injectate types used for PIT was reported by Buntragulpoontawee et al., with a literature review through August 2020 [24]. To date, a systematic review pertaining to both ultrasound-guided PIT (UPIT) and ultrasound-guided percutaneous CTR (UPCTR) has not been reported. Therefore, we performed this systematic review based on published studies to examine the quality of evidence, effectiveness, and safety of UPIT and UPCTR in CTS. We hypothesized that both ultrasound-guided interventions are safe and effective for CTS.

## 2. Materials and Methods

This review conforms to the Preferred Reporting Items for Systematic Reviews and Meta-Analyses (PRISMA) statement [25]. The protocol was registered at PROSPERO 2022 CRD42022384170 (available from https://www.crd.york.ac.uk/prospero/display_record.php?ID=CRD42022384170; accessed on 29 December 2022).

### 2.1. Data Sources and Search Strategy

Four databases (i.e., PubMed, EMBASE, Scopus, and Cochrane) were systematically searched for the relevant literature from inception up to 30 June 2022. The search strategy included a combination of the following search terms together using logical Boolean operators “AND” and “OR”: “ultrasound-guided,“ “hydrodissection,“ “intervention,“ “injection,“ “surgery,“ “release,“ “complication,“ “adverse effect,“ and “CTS.“ The references of the articles were manually searched to identify additional relevant publications.

### 2.2. Inclusion and Exclusion Criteria

Inclusion criteria were as follows: (1) English language clinical trials, retrospective studies, or case series in humans assessing the efficacy and safety of either UPIT or UPCTR for CTS; (2) studies with outcome measures including changes in pain intensity, other clinical symptoms, function, electrophysiological measures, day of return to work, or the cross-sectional area (CSA) of the MN; and (3) UPCTR studies requiring the release of the entire FR width. There were no limitations on the therapy provided to the control groups for included studies. Exclusion criteria were as follows: (1) studies using a mixed injectate for UPIT, and (2) studies with components of open surgery or endoscopic release for UPCTR.

### 2.3. Data Extraction

Two reviewers (KHSL and YTW), each with more than seven years of research experience, independently performed full-text reviews to extract data for analysis. The collected data included study the design, inclusion criteria, ultrasound-guided (UG) intervention and treatment allocation, participants’ characteristics, CTS severity, outcome measurements, follow-up period, and safety outcomes (Figure 1). Discrepancies in study selection were resolved by a third reviewer (KDR).

### 2.4. Outcome Measures for Meta-Analysis

The primary outcomes of interest for meta-analysis were changes in symptom severity, measured using a visual analogue scale (VAS), numerical rating scale (NRS), or the Boston Carpal Tunnel Questionnaire symptom scale (BCTQ-SS) and function scale (BCTQ-FS) [26]. VAS or NRS values were converted to a 0–10 format for analysis, and BCTQ values were converted to a standard 0–5 format for symptom and function scale analyses.

### 2.5. Bias Assessment

Bias risks and the critical appraisal of the manuscripts were independently assessed using the Cochrane risk-of-bias tool for randomized trials, version 2 (RoB 2) [27] and the Joanna Biggs Institute (JBI) Critical Appraisal Checklist for Case Series [28] and Cohort Studies [29]. Using the RoB 2 tool, an overall risk of bias for a specific outcome was judged as “low risk ” only when all individual domains were scored as low risk, “some concerns” if any one category was scored as “some concern,” and “high risk” if any one category was scored as “high risk” or with more than one category scored as “some concern” [27]. The overall bias risk for case series or cohort studies was scored as “low” if more than 7 of 10 items or 8 of 11 items on the JBI Critical Appraisal Checklist for Cases Series [28] or Cohort Studies [29], respectively, were scored as low risk. The risk of bias was assessed by two independent reviewers (KHSL and AEA), with discrepancies resolved by a third reviewer (KDR).

### 2.6. Analysis

All meta-analyses were conducted using Revman version 5.4.1 [30]. A random effects model was used to pool study results. Changed scores for continuous outcome measures (VAS, BCTQ-SS, or BTCT-SS) were pooled as standardized mean differences with a confidence interval of 95% and a weighted mean difference was calculated. Potential clinical importance was interpreted according to minimal clinically important differences for VAS and BCTQ [31,32]. Heterogeneity between studies was reported as an I^2^ value and an overall effect was reported as a Z score with a corresponding *p* value.

## 3. Results

### 3.1. Study Selection and Characteristics

#### Selection (Figure 1)

After our primary search, 84 potentially relevant studies on UPIT were identified, of which 36 publications met the inclusion criteria. Among them, 18 studies reported outcomes of UPIT with corticosteroid injection, six with 5% dextrose in water (D5W), seven with platelet-rich plasma (PRP), two with hyaluronidase, one with hyaluronic acid (HA), one with insulin, and one with ozone. The publication types included 31 randomized controlled trials (RCTs), three retrospective studies, and one pilot study. Regarding UPCTR, 90 potentially relevant papers were identified, of which 24 met the inclusion criteria, including four RCTs, four cohort studies, and 16 case series.

### 3.2. Bias Analysis of UPIT and UPCTR Studies

#### 3.2.1. Bias Analysis of RCTs (Table 1)

The bias assessments for all the RCTs are listed in Table 1. Of 30 RCTs of UPIT, 17 showed a high overall bias risk, mainly due to high bias owing to deviations from intended interventions because the study participants could not be blinded; four showed some concerns in overall bias risk; and nine had a low overall risk of bias.

**Table 1 diagnostics-13-01138-t001:** Bias table for RCTs of UPIT and UPCTR.

Author	Domain 1	Domain 2	Domain 3	Domain 4	Domain 5	Overall Bias
Bias table for RCTs of UPIT using Steroids
Ustun (2013) [33]						
Lee (2014) [34]						
Makhlouf (2014) [35]						
Eslamian (2017) [36]						
Karaahmet (2017) [37]						
Wang (2017) [38]						
Chen (2018) [39]						
Ba-baei-Ghazani (2018) [40]						
Roghani (2018) [41]						
Roh (2019) [42]						
Rayegani (2019) [43]						
Hsu (2020) [44]						
Ba-baei-Ghazani (2020) [45]						
Mezian (2021) [46]						
Wang (2021) [47]						
Mathew (2022) [48]						
Bias table for RCTs of UPIT using 5% Dextrose
Wu (2017) [49]						
Wu (2018) [50]						
Lin (2020) [51]						
Lin (2021) [52]						
Bias table for RCTs of UPIT using Platelet-Rich Plasma
Wu et al. (2017) [53]						
Malahias et al. (2018) [54]						
Senna et al. (2019) [55]						
Shen 2019 [56]						
Chen et al. (2021) [57]						
Bias table for RCTs of UPIT using other Injectates
Su et al. (2021) [58]						
Alsaeid et al. (2019) [59]						
Elawa-my et al. (2020) [60]						
Kamel et al. (2021) [61]						
Forogh et al. (2021) [62]						
Bias table for RCTs of UPCTR
Capa Grasa (2014) [63]						
Rojo-Manaute (2016) [64]						
Zhang (2019) [65]						
Fuente (2021) [66]						

Domain 1, bias arising from the randomization process; Domain 2, bias due to deviations from intended interventions; Domain 3, bias due to missing outcome data; Domain 4, bias in the measurement of the outcome; and Domain 5, bias in the selection of the reported result. Red color dots signifie high bias, orange color dots point to some concerns on bias analysis, and green color dots denote low bias.

#### 3.2.2. Bias Analysis of Cohort Studies

The critical appraisals for cohort studies are summarized in Table 2. Only one cohort study showed a high bias risk.

#### 3.2.3. Bias Analysis of Case Series (Table 3)

The critical appraisals for case series studies are presented in Table 3. The case series for the UPIT had a low-risk bias; however, five of 16 case series studies for UPCTR had a high-risk bias and the rest of the studies had a low-risk bias.

**Table 3 diagnostics-13-01138-t003:** JBI Critical Appraisal Checklist for Case Series.

	Criteria and Corresponding Scores												
Author	#1	#2	#3	#4	#5	#6	#7	#8	#9	#10	Total	%	Bias Risk
Bias table for case series of UPIT using Dextrose													
Li et al. (2021) [73]	1	1	1	1	1	1	1	1	1	1	10	100	Low
Chao et al. (2022) [74]	1	1	1	1	1	1	1	1	1	1	10	100	Low
Bias table for case series of UPIT using Platelet-Rich Plasma													
Malahias et al. (2015) [75]	1	1	1	1	1	0	1	1	1	1	9	90	Low
Bias table for case series for UPCTR													
Chern et al. (2015) [76]	1	1	1	1	1	1	1	1	0	U	8	80	Low
Guo et al. (2015) [77]	U	1	1	U	U	1	0	1	0	U	4	40	High
Guo et al. (2017) [78]	1	1	1	1	1	1	1	1	0	1	9	90	Low
Petrover et al. (2017) [79]	1	1	1	1	1	1	0	1	0	1	8	80	Low
Henning et al. (2018) [80]	1	1	1	U	U	1	0	1	0	1	6	60	High
Luanchumroen et al. (2019) [81]	1	1	1	1	1	1	0	1	0	1	8	80	Low
Wang et al. (2019) [82]	1	1	1	1	1	1	1	1	0	1	9	90	Low
Chappell et al. (2020) [83]	1	1	1	1	1	1	1	1	0	U	8	80	Low
Hebbard et al. (2020) [84]	0	U	U	U	U	1	0	1	0	1	3	30	High
Joseph et al. (2020) [85]	1	1	1	1	1	1	1	1	0	1	9	90	Low
Kamel et al. (2020) [86]	1	1	1	1	1	1	1	1	0	1	9	90	Low
Wang et al. (2021) [87]	1	1	1	1	1	1	0	1	0	U	7	70	High
Leiby et al. (2021) [88]	1	1	1	1	1	1	1	1	0	1	9	90	Low
Loizides (2021) [89]	1	1	1	1	1	1	1	1	0	U	8	80	Low
Lee (2022) [90]	1	1	1	1	0	1	1	1	0	1	8	80	Low
Fowler (2022) [91]	1	1	1	0	U	1	1	1	0	1	7	70	High
Quality measures of case series studies based on the following listed criteria:													
#1. Were there clear criteria for inclusion in the case series?													
#2. Was the condition measured in a standard, reliable way for all participants included in the case series?													
#3. Were valid methods used for identification of the condition for all participants included in the case series?													
#4. Did the case series have consecutive inclusion of participants?													
#5. Did the case series have complete inclusion of participants?													
#6. Was there clear reporting of the demographics of the participants in the study?													
#7. Was there clear reporting of clinical information of the participants?													
#8. Were the outcomes or follow-up results of cases clearly reported?													
#9. Was there clear reporting of the demographic information of the presenting site(s)/clinic(s)?													
#10. Was statistical analysis appropriate?													

NB: 1 indicates the article does fulfill the specified criteria; 0 indicates the article does not fulfill the stated criteria; U indicates the article is unclear about the criteria; Red color front signifies high bias, and green color words denote low bias.

### 3.3. UPIT Study Characteristics

#### 3.3.1. UPIT Using Corticosteroids

Among the 18 studies on the outcomes of UPIT with corticosteroid injection (Table 4), eight compared the efficacy of ultrasound-guided injections vs. that of blind landmark-guided injections [33,34,35,36,37,39,42,43]. Several recent UPIT studies investigated the efficacy of different concentrations of the same steroid (four UPIT studies [37,39,41,44,47]), use of different steroids (one study [48]), hydrodissection superficial vs. deep to the MN (one study [40]), effects of short vs. long-axis UPIT (one study [43]), intra- vs. extraepineurial UPIT (one study [67]), ulnar vs. radial approaches for UPIT (one study [45]), and perineural vs. peritendinous approaches (one study [46]).

#### 3.3.2. UPIT with D5W

Among the six studies of UPIT using D5W (Table 5), four were double-blind RCTs [49,50,51,52], one investigated UPIT with D5W vs. normal saline [49], one evaluated UPIT with D5W vs. corticosteroids [50], and two compared different volumes of D5W in UPIT [51,52]. Of the two retrospective case series [73,74], one reported the long-term outcomes of UPIT with D5W [73] and another evaluated the effectiveness of UPIT with D5W in the post-surgical persistence and recurrence of CTS [74].

#### 3.3.3. UPIT with PRP

Seven studies evaluated the effectiveness UPIT with PRP [53,54,55,56,57,69,75] (Table 6). Of these studies, one RCT and one case study compared PRP with splint use [53,69], two RCTs compared PRP with normal saline [54,57], one RCT compared PRP with steroids [55], and one RCT compared PRP with D5W [56].

#### 3.3.4. UPT with Other Injectates

Of the five other RCTs of UPIT using other injectates [58,59,60,61,62] (Table 7), one compared HA with normal saline [58]; one evaluated hyaluronidase vs. steroids [59]; one investigated hyaluronidase vs. normal saline [60]; one compared insulin alone vs. steroids alone, steroids alone, and steroids followed by insulin [61]; and one evaluated ozone vs. steroids [62].

### 3.4. UPCTR Study Characteristics

Of the 24 included studies for UPCTR, (Table 8) there were four RCTs [63,64,65,66]: two investigated UPCTR using a hook knife vs. mini-OCTR [63,64]; one compared miniscalpel needles plus steroid injection with corticosteroid injection only [65]; and one evaluated a U-shaped probe/trough plus 5 mm Dovetail blades vs. OCTR [66]. There were four case-cohort UPCTR studies [17,70,71,72]: one compared UPCTR with mini-OCTR [17]; one investigated UPCTR vs. OCTR [72]; one evaluated UPCTR using a 22 G hypodermic needle vs. UPCTR plus corticosteroid injection [70]; and the last case-cohort study compared the use of a UPCTR of an uncoated multifilament stainless steel wire looped thread with no interventions [71]. Among 16 case series [76,77,78,79,80,81,82,83,84,85,86,87,88,89,90,91], six investigated the effects of different hook knives [76,79,81,82,87,89], five evaluated the use of microknives [80,83,85,86,88], two analyzed loop threads [77,78], one assessed a microblade [84], and one investigated an 18 G needle with a tip bent in the opposite direction to the needle bevel [90].

### 3.5. Meta-Analysis Results

#### 3.5.1. UPIT with Steroids vs. Landmark-Guided Steroid Injection (Figure 2)

Steroid injection by ultrasound guidance reduced symptoms significantly more than landmark guidance in the pooled results with variable heterogeneity, as indicated by improvement in VAS0-10 (MD: −1.21 [95% CI: −2.05 to −0.37]; *p* = 0.005) and BCTQ-SS (MD: −0.35 [95% CI: −0.66 to −0.05]; *p* = 0.02). Functional improvement (BCTQ-SF) was significantly greater after ultrasound-guided steroid injection (MD: −0.26 [95% CI: −0.51 to −0.00] *p* = 0.05) with low heterogeneity (I^2^ = 34%).

**Figure 2 diagnostics-13-01138-f002:**
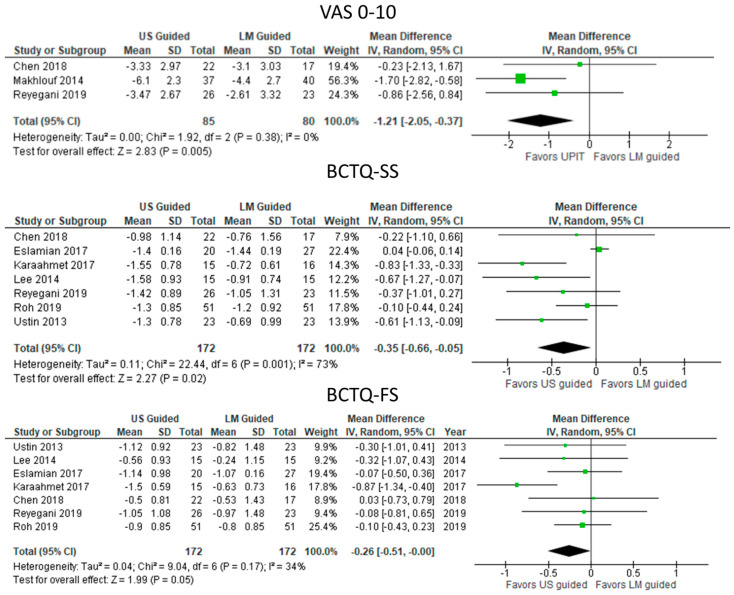
Meta—analysis of UPIT with steroids vs. landmark-guided steroid injection. US, ultrasound; LM, landmark; VAS, visual analogue scale; BCTQ—SS, Boston Carpal Tunnel Questionnaire—symptom scale; BCTQ—FS, Boston Carpal Tunnel Questionnaire—functional scale. The green squares represent the differences between the means of each of the two groups compared, i.e., the mean differences. The size of the green square represents how much that individual study affects the overall outcome of the meta—analysis, i.e., the weight of the studies on the meta—analysis. The precision of the study depends on the 95% confidence interval for that mean difference, i.e., the length of the straight lines, the shorter the lines, the more precise the mean differences. The black trapezoid is the pooled mean difference of all the studies combined in consideration of the weight of each study, with its confidence interval extending from the left tip to the right tip.

#### 3.5.2. UPIT with High-Dose vs. Low-Dose Steroids (Figure 3)

Higher doses of steroid injection by ultrasound guidance reduced symptoms significantly more than lower doses of steroids in the pooled results with low heterogeneity (I^2^ = 0%), as indicated by improvement in VAS0-10 (MD: −0.020 [95% CI: −0.38 to −0.02]; *p* = 0.03) and BCTQ-SS (MD: −0.22 [95% CI: −0.28 to −0.15]; *p* < 0.00001). Functional improvement (BCTQ-SF) was significantly greater after ultrasound guidance (MD: −0.06 [95% CI: −0.10 to −0.02] *p* = 0.004).

**Figure 3 diagnostics-13-01138-f003:**
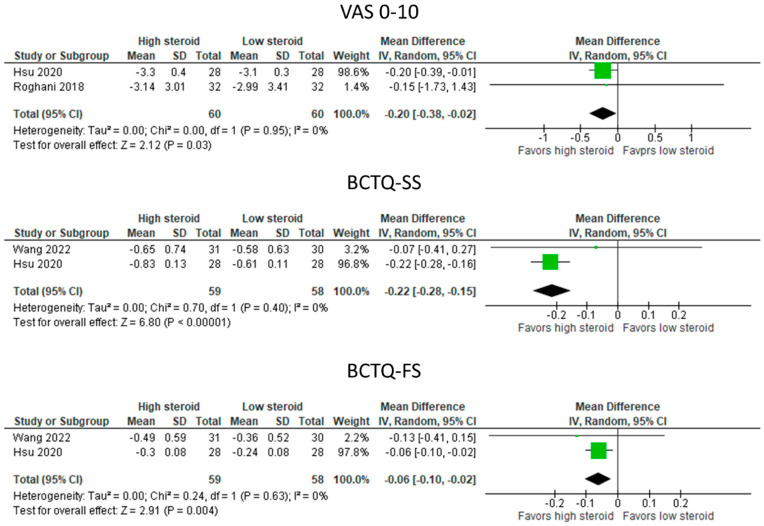
Meta—analysis of UPIT with high—dose vs. low—dose steroids. VAS, visual analogue scale; BCTQ—SS, Boston Carpal Tunnel Questionnaire—symptom scale; BCTQ—FS, Boston Carpal Tunnel Questionnaire—functional scale. The green squares represent the differences between the means of each of the two groups compared, i.e., the mean differences. The size of the green square represents how much that individual study affects the overall outcome of the meta—analysis, i.e., the weight of the studies on the meta—analysis. The precision of the study depends on the 95% confidence interval for that mean difference, i.e., the length of the straight lines, the shorter the lines, the more precise the mean differences. The black trapezoid is the pooled mean difference of all the studies combined in consideration of the weight of each study, with its confidence interval extending from the left tip to the right tip.

#### 3.5.3. UPIT with D5W vs. Control Injections (Figure 4)

UPIT with D5W reduced symptoms significantly more than UPIT with either triamcinolone (Wu, 2018) or saline (Wu, 2017-1) in the pooled results with moderate to high heterogeneity, as indicated by improvement in VAS0-10 (MD: −0.82 [95% CI: −1.64 to 0.01]; *p* = 0.05) and BCTS-SS (MD: −0.41 [95% CI: −0.50 to −0.31]; *p* < 0.00001). Functional improvements were significantly greater with UPIT using D5W as the injectate (MD: −0.55 [95% CI: −0.88 to −0.33]; *p* = 0.0008), with high study heterogeneity (I^2^ = 97%).

**Figure 4 diagnostics-13-01138-f004:**
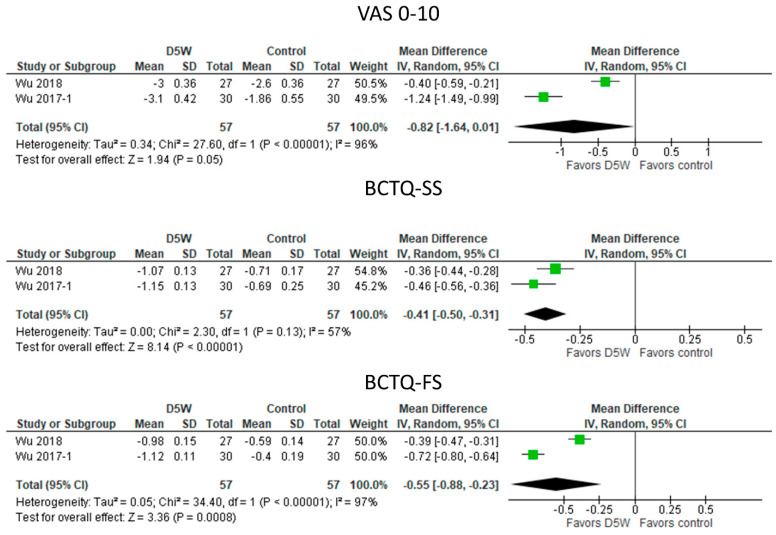
Meta—analysis of UPIT with D5W vs. control injections. VAS, visual analogue scale; BCTQ—SS, Boston Carpal Tunnel Questionnaire—symptom scale; BCTQ—FS, Boston Carpal Tunnel Questionnaire—functional scale. The green squares represent the differences between the means of each of the two groups compared, i.e., the mean differences. The size of the green square represents how much that individual study affects the overall outcome of the meta—analysis, i.e., the weight of the studies on the meta—analysis. The precision of the study depends on the 95% confidence interval for that mean difference, i.e., the length of the straight lines, the shorter the lines, the more precise the mean differences. The black trapezoid is the pooled mean difference of all the studies combined in consideration of the weight of each study, with its confidence interval extending from the left tip to the right tip.

#### 3.5.4. UPIT with Higher Volumes of D5W vs. Lower Volumes of D5W (Figure 5)

Data for BCTS subscales were unavailable. Symptom improvements, however, were significantly greater with higher volumes of D5W than with lower volumes of D5W (4 mL vs. 1 mL D5W; MD: −2.21 [95% CI: −3.19 to −1.23]; *p* < 0.00001).

**Figure 5 diagnostics-13-01138-f005:**
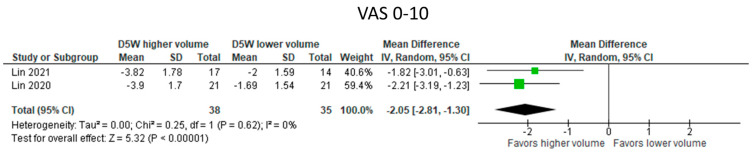
Meta—analysis of UPIT with higher volumes of D5W vs. lower volumes of D5W. VAS, visual analogue scale. The green squares represent the differences between the means of each of the two groups compared, i.e., the mean differences. The size of the green square represents how much that individual study affects the overall outcome of the meta—analysis, i.e., the weight of the studies on the meta—analysis. The precision of the study depends on the 95% confidence interval for that mean difference, i.e., the length of the straight lines, the shorter the lines, the more precise the mean differences. The black trapezoid is the pooled mean difference of all the studies combined in consideration of the weight of each study, with its confidence interval extending from the left tip to the right tip.

#### 3.5.5. UPIT with PRP vs. Control Treatments (Figure 6)

UPIT with PRP did not significantly reduce VAS0-10 compared with the control treatment. However, the reduction in symptoms, as measured by BCTQ-SS, was significantly greater after UPIT with PRP (MD: −0.36 [95% CI −0.43 to −0.30]; *p* < 0.00001). In addition, functional improvement also favored treatment with PRP injection (MD: −0.29 [95% CI: −0.47 to −0.12]; *p* = 0.001).

**Figure 6 diagnostics-13-01138-f006:**
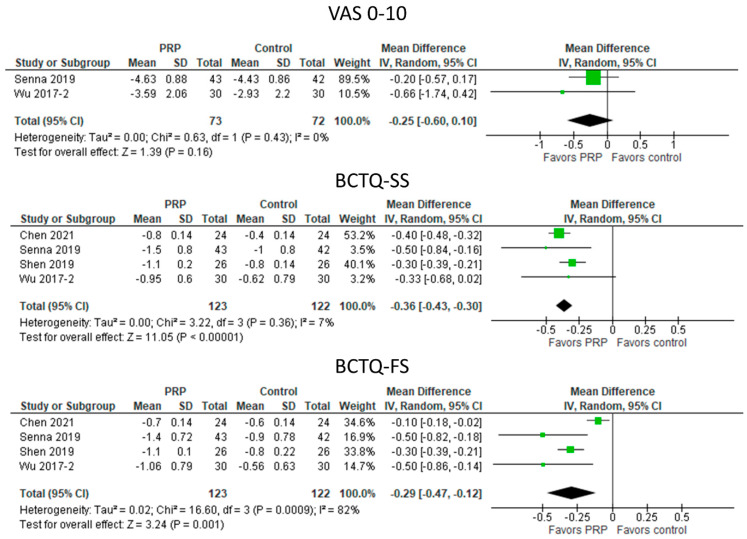
Meta—analysis of UPIT with PRP vs. control treatments. VAS, visual analogue scale; BCTQ—SS, Boston Carpal Tunnel Questionnaire—symptom scale; BCTQ—FS, Boston Carpal Tunnel Questionnaire—functional scale. The green squares represent the differences between the means of each of the two groups compared, i.e., the mean differences. The size of the green square represents how much that individual study affects the overall outcome of the meta—analysis, i.e., the weight of the studies on the meta—analysis. The precision of the study depends on the 95% confidence interval for that mean difference, i.e., the length of the straight lines, the shorter the lines, the more precise the mean differences. The black trapezoid is the pooled mean difference of all the studies combined in consideration of the weight of each study, with its confidence interval extending from the left tip to the right tip.

#### 3.5.6. UPCTR Vs. Surgery (Figure 7)

Comparative VAS0-10 data were unavailable. UPCTR outperformed open surgery in terms of symptom improvement (MD: −0.40 [95% CI: −0.70 to −0.10]; *p* = 0.009), but functional improvements were not significantly different between UPCTR and open surgery.

**Figure 7 diagnostics-13-01138-f007:**
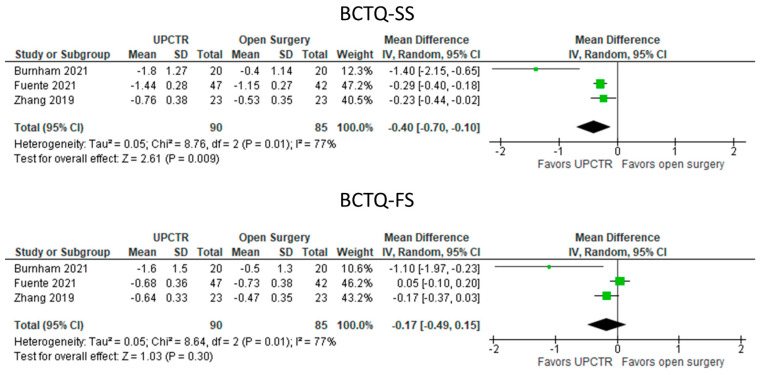
Meta—analysis of UPCTR vs. surgery. BCTQ—SS, The Boston Carpal Tunnel Questionnaire—symptom scale; BCTQ—FS, The Boston Carpal Tunnel Questionnaire—functional scale. The green squares represent the differences between the means of each of the two groups compared, i.e., the mean differences. The size of the green square represents how much that individual study affects the overall outcome of the meta—analysis, i.e., the weight of the studies on the meta—analysis. The precision of the study depends on the 95% confidence interval for that mean difference, i.e., the length of the straight lines, the shorter the lines, the more precise the mean differences. The black trapezoid is the pooled mean difference of all the studies combined in consideration of the weight of each study, with its confidence interval extending from the left tip to the right tip.

## 4. Discussion

### 4.1. Meta-Analysis Findings

We found that UPIT with steroids significantly improved symptoms and function in carpal tunnel syndrome compared with landmark guidance and UPIT with high-dose steroids was superior to that with low-dose steroids. Meta-analyses on UPIT with D5W or PRP showed that these treatments outperformed their corresponding controls, including steroid injections. Further investigations are needed to determine the relative cost-effectiveness of D5W vs. PRP injections for CTS. Meta-analyses of the effects of UPCTR vs. open surgery revealed that UPCTR was at least as efficacious as open surgery and performed better in terms of symptom reduction, but not functional improvement.

### 4.2. Literature Review of UPIT

The mechanisms of UPIT for CTS can be either mechanical (hydrodissection) or pharmacological effects.

#### 4.2.1. Mechanical Effect (Hydrodissection)

Hydrodissection can extricate the entrapped MN from the surrounding adhesive/compressive tissue by exerting a non-specific effect of fluid-under-force to further increase the blood flow and alleviate nerve compression injury [92,93,94,95]. Extrication with an associated restoration of MN kinematics breaks the vicious cycle of compression, inflammation, SSCT fibrosis, and nerve ischemia and could contribute to subsequent nerve regeneration [49,53,92,95]. Evers et al. [96] reported that hydrodissection with normal saline (NS) reduces MN gliding resistance within the carpal tunnel in the cadaveric wrist. This effect cannot be explained based on the fluid lubrication effect, as it persists without degradation over 1000 repetitions. Wu et al. [93] confirmed the clinical benefits of hydrodissection with NS for mild-to-moderate CTS. A single injection of 5 mL NS resulted in medium-term (6-month) symptom reduction and decreased MN CSA. In addition, a study comparing three different injectate volumes (1, 2, or 4 mL) showed that hydrodissection with a higher volume of injectate yielded better nerve mobility and decreased MN CSA [52]. Simultaneous hydrodissection above and below the MN was more effective than only hydrodissection between the FR and MN [97]. The minimum volume requirement for an optimum mechanical effect and whether this effect includes a beneficial effect from the NS require further investigations.

#### 4.2.2. UPIT Results by Injectate

##### Corticosteroids (Table 1)

In recent decades, corticosteroids have been the most frequently used ultrasound-guided injectate for CTS. The proposed primary mechanism of corticosteroid injection benefit in CTS is a reduction in intracarpal pressure via exertion of an anti-inflammatory effect, decompressing the nervi nervorum, rather than a direct pharmacological effect on the MN [98]. Various ultrasound-guided techniques for CTS treatment by corticosteroid injection have been reported, with inconclusive results for comparative effectiveness but intriguing preliminary findings. No significant differences were reported in the subjective and objective measurements between perineural and peritendinous (between carpal bone) corticosteroid injections through 12 weeks of follow-up [46]. Babaei-Ghazani et al. [40] concluded that ultrasound-guided corticosteroid injection above or below the MN was equally effective in symptom reduction, functional improvement, and electrophysiologic and sonographic findings. Hsu et al. [67] reported that intraepineurial corticosteroid injections outperformed extraepineurial injections in improving patient satisfaction, symptom relief, and MN CSA. Lee et al. [34] showed that a short-axis in-plane approach below and above the MN improved the symptoms, function, CSA, and electrophysiological parameters more than a short-axis out-of-plane approach below the MN. Rayegani et al. [43] demonstrated that a long-axis in-plane approach above the MN tended to decrease CSA more than did a short-axis in-plane approach below the MN; however, the between-group difference did not reach significance. Babaei-Ghazani et al. [45] revealed no differences between ultrasound-guided short-axis ulnar and radial injection direction approaches for CTS. Wang et al. [47] showed that ultrasound-guided hydrodissection using triamcinolone or corticosteroid perineural injection alone resulted in clinical and electrophysiological improvement in patients with CTS, but hydrodissection did not offer additional benefits. However, the hydrodissection techniques in Wang et al.’s study have been considered imperfect [99], and the volume and concentration of steroids used for hydrodissection and perineural injection were different, which might be interventional confounding variables [99]. Although a few studies showed similar efficacy between landmark-based and ultrasound-guided injection approaches [36,42], most studies demonstrated that ultrasound-guided injection resulted in significantly greater clinical improvements [33,35,37,39].

The pharmacological effects of corticosteroids may outweigh those of injection techniques, partially explaining the diversity of findings in the studies mentioned above [100]. Moreover, the hydrodissection effects might have been less notable in corticosteroid injection studies as most injection methods used only 1–3 mL of corticosteroids, which might not have been sufficient for an adequate hydrodissection effect. However, Wang et al. [47] conducted a single-blind trial in which the participants were randomly assigned hydrodissection with a mixture of 1 mL of triamcinolone acetonide (10 mg/mL), 1 mL of 2% lidocaine, and 8 mL of NS or perineural injection with 1 mL of triamcinolone acetonide (10 mg/mL) and 1 mL of 2% lidocaine. They reported no additional benefit from the corticosteroid injection of a 10 mL volume compared with that from the injection of a 2 mL volume with the same corticosteroid dosage. This finding suggests that the anti-inflammatory effect of corticosteroids is more important than the hydrodissection effect. However, this conclusion is weakened by a potential bias from a difference in corticosteroid concentration through dilution and the limitation in their techniques of hydrodissection which may have affected the outcome [99].

The follow-up periods of the corticosteroid injection studies varied, with therapeutic benefits reported for 3 months by Lee et al. [34], Üstün et al. [33], and Wang et al. [38] and 6 months and 16 months by Makhlouf et al. [35] and Yeom et al. [68], respectively. Notably, recent trials demonstrated no dose-dependent effect of ultrasound-guided corticosteroid injections [41,44]. Although no difference in clinical outcomes between particulate (triamcinolone acetonide) and non-particulate (dexamethasone sodium phosphate) corticosteroid injections for CTS was observed, the particulate group showed significantly longer post-injection pain duration [48].

The efficacy of ultrasound-guided corticosteroid injection for CTS is uncertain because of the absence of a well-designed control group. The published research only compared different guided methods, i.e., the ultrasound-guided method vs. blind injection or different ultrasound-guided techniques. These published studies might have overestimated the therapeutic effect owing to the absence of a well-controlled placebo group. A Cochrane review concluded that the beneficial effect of corticosteroid injections using a blind technique has only a short-term benefit compared with that of a placebo injection (about one month) [101]. This finding suggests that the medium- and long-term clinical benefits reported represent, in part, a placebo effect. The possible adverse effects of corticosteroids include widespread axonal and myelin degeneration, skin thinning, tendon rupture, soft tissue atrophy, steroid flare, crystal-induced synovitis, and hot flushes [102,103]. Additional randomized, double-blind, controlled trials with well-designed control groups with limited therapeutic activity, such as those administered NS, or active non-steroid treatment comparison groups, are needed to confirm the clinical benefit of ultrasound-guided corticosteroid injection and its comparative risk/benefit ratios.

##### Five Percent Dextrose in Water (Table 2)

In 2017, Wu et al. [49] first proved that a single hydrodissection with 5 mL D5W, compared with NS hydrodissection, could significantly improve the symptoms, electrophysiological measures, and MN CSA for CTS. Wu et al. [50] subsequently reported the waning benefits of single corticosteroid hydrodissection from 3–6 months in contrast with cumulative improvement in the subjective symptoms and disability associated with single hydrodissection with 5 mL D5W. Lin et al. [51] showed that increasing the volume of D5W showed better efficacy in reducing CTS-related symptoms and disability in a 24-week post-injection follow-up. A higher injectate volume also increased nerve mobility and reduced nerve swelling [52].

Li et al. [73] administered UPIT using D5W with multiple injection sections (mean 2.2 injections with 10 mL D5W), with 1.7 to 3 injections, to achieve an effective outcome over a mean of 15.8 months follow-up, and they found that only 1% (2/185) of the patients ultimately underwent surgery [73]. Given the early level A evidence that hydrodissection with D5W for CTS is likely to outperform or supplement the current conservative treatment approaches and significantly reduce the probability of surgery, this injection is listed as one of the treatment options by UpToDate [104]. In addition, the 20th edition of Harrison’s Principles of Internal Medicine also officially lists this method as an alternative treatment for CTS [105]. Recently, a retrospective study by Chao et al. [74] suggested that UPIT with D5W is clinically important with a durable effect in patients with failure of CTS surgery. They retrospectively followed 36 patients with persistent or recurrent symptoms after CTS surgery for a mean post-injection follow-up of 33 months and symptom relief of more than 50% was reported by 22 patients (61%) after a mean of 3.1 injections of UPIT with 10 mL D5W.

The pharmacological effects of D5W for CTS are still unclear. Theoretically, they include stabilization effects on neural activity, normalization of glucose metabolism, and a decrease in neurogenic inflammation, reducing neuropathic pain via multifactorial mechanisms. Glucose has been speculated to indirectly inhibit capsaicin-sensitive receptors (e.g., transient receptor potential vanilloid receptor-1) and block the secretion of substance P and calcitonin gene-related peptides, which are pro-nociceptive substances involved in neurogenic inflammation [95,106,107,108,109,110,111]. Wu et al. [112] observed that high glucose concentrations could mitigate TNF-α-induced NF-κB activation, upregulation of pro-inflammatory cytokines, and metabolic dysfunction in nerve cells; the in vitro findings support the hypothesized mechanism that glucose may reduce neurogenic inflammation. Moreover, pre-treatment with TNF-α also promotes energy uptake, leading to glucose deprivation [113,114,115,116,117], and glucose exposure could restore impaired glucose metabolism. Additional studies are needed to further explore the apparent ameliorative pharmacological mechanisms of D5W in CTS and its optimal dosage and frequency for CTS injection.

##### PRP (Table 3)

PRP is rich in various growth factors that can help neural repair and regeneration, as shown by animal and in vitro studies, but PRP must be used at patients’ own expense [118]. Additionally, PRP can reduce the levels of pro-inflammatory cytokines, local inflammation, and intracarpal pressure to normalize the pressure on MN [119]. Since 2015, clinical studies have investigated the efficacy of UPIT with PRP for CTS [53,54,55,57,69,75]. Wu et al. [53] showed that ultrasound-guided PRP injection was more effective for symptom relief and CSA of MN than using a wrist splint for six months. Malahias et al. [54] performed the first double-blind placebo-controlled PRP vs. NS injection trial and found significant between-group differences favoring PRP at the 3-month follow-up. Senna et al. [55] conducted a randomized, single-blind controlled trial and reported that PRP was more effective in reducing the symptoms and improving electrophysiologic measures at three months of post-injection than corticosteroid injection. A non-blind trial with a short-term follow-up (4 weeks) reported no additional benefit from the addition of 1 mL of PRP to the use of a wrist splint for CTS compared with the use of a splint alone. However, that trial had a very short follow-up and the volume used was not expected to produce a hydrodissection effect [69]. Shen et al. [56] compared PRP to D5W administration with a single injection only for moderate CTS. They found more benefits from PRP injection in functional improvement, electrophysiologic measures, and CSA of MN.

In 2021, Chen et al. [57] first demonstrated and compared the long-term efficacy (1-year follow-up) of a single PRP injection vs. NS in terms of symptom relief, functional scores, and MN CSA in a randomized, double-blind, controlled trial. Recently, Lai et al. [120] retrospectively followed up on 81 patients at least two years after a single PRP injection; after a mean of 43.8 months (24–60 months) post-injection follow-up, 70% of patients reported symptom relief >50%. Considering the biological properties of PRP, it is possibly the most effective injectate at present, especially for more severe-grade presurgical CTS. However, further studies with larger sample sizes and comparing PRP with other active injectates for CTS, considering differences in dosage needs and cost efficacy, are needed.

##### HA, Hyaluronidase, Insulin, or Ozone (Table 4)

Single studies with a small sample size have reported results from the ultrasound-guided injection of HA, hyaluronidase, insulin, or ozone for mild-to-moderate CTS. Studies with larger sample sizes and more extended follow-up periods are needed in the future to evaluate their efficacy or comparative efficacy.

HA

HA, which has anti-adhesion effects, has clinical applications for post-surgical adhesion [121,122]. Su et al. [58] reported that compared with an NS injection, a single ultrasound-guided HA injection had significant efficacy for symptom and functional improvement at two weeks post-injection. Retention of the majority of the HA injectate was still visualizable by ultrasound one hour after injection, while NS was almost completely absorbed. Prolonged HA retention surrounding MN and its anti-adhesion effect may contribute to early symptom relief through lubrication, MN mobility improvement, and decreased pressure within the carpal tunnel [123,124,125].

Hyaluronidase

Hyaluronidase, an enzyme catalyzing the hydrolysis of HA, can reduce viscoelasticity, increase tissue permeability, and allow the local anesthetic to diffuse through the surrounding tissue [126]. Several studies have revealed significant but brief pain reduction from the perineural injection of hyaluronidase in a local anesthetic solution vs. local anesthetic only in symptomatic CTS [126,127,128]. A randomized control trial with mid-term follow-up was conducted to compare hydrodissection under the ultrasound guidance of hyaluronidase vs. dexamethasone for mild-to-moderate CTS [59]; this trial demonstrated significantly greater improvements in the symptoms, function, electrophysiological findings, and CSA of MN in a 6-month follow-up in the hyaluronidase group compared with those in the dexamethasone group [59]. In addition, in 2020, Elawamy et al. [60] reported that ultrasound-guided injection with hyaluronidase combined with NS, compared with NS alone, resulted in significantly more improvements in the pain, function, electrophysiological parameters, and CSA of MN in a 6-month follow-up for mild-to-moderate CTS. The possible mechanism may be that HA accumulates around demyelinated nerves, inhibiting oligodendrocyte precursor cell maturation and remyelination, and injected hyaluronidases can stimulate remyelination via hydrolysis of local HA [60,129].

Insulin

Kamel et al. [61] reported that ultrasound-guided injection with corticosteroids plus insulin, compared with corticosteroids alone, significantly decreased the CSA of MN. However, symptoms and functional outcomes did not differ between the groups. Further research on the use of insulin for injection is anticipated based on the potential role of insulin (e.g., acting like a growth factor) to determine its therapeutic efficacy in compression-related nerve dysfunction [130].

Ozone

Ozone exerts anti-inflammatory and analgesic properties by inhibiting pro-inflammatory mediators [131]. Forogh et al. [62] compared ultrasound-guided injection with ozone vs. corticosteroid for mild-to-moderate CTS in a randomized trial; ozone was non-inferior to corticosteroid injection with respect to pain reduction and functional improvement at the 12-week follow-up, although improvement in electrophysiological parameters and MN CSA was observed only among patients after corticosteroid injection.

### 4.3. Ultrasound-Guided Percutaneous Carpal Tunnel Release (Table 5)

Studies have confirmed that UPCTR is feasible and it is receiving increasing attention in the literature [21,64]. Compared with the standard-of-care OCTR, UPCTR has the following advantages:

(1) smaller incision size, typically 0.1–0.3 cm only;

(2) better safety due to full-time continuous visualization and monitoring of the neurovascular structures and the instruments while releasing the FR [21];

(3) significantly faster wound healing and resumption of daily activities and work [21], with five studies reporting a return to work as early as the first week post-UPCTR [64,77,78,84];

(4) faster improvement in the short-term functional scores, grip strength, and paresthesia disappearance [21]; and

(5) non-inferior median and long-term outcomes when compared with those of the current standard-of-care OCTR.

Due to the surgery’s minimally invasive nature and much faster healing, five of the included 24 studies reported simultaneous bilateral UPCTR [63,83,84,85,88], which is not feasible when performing mini-OCTR. Early studies have suggested that UPCTR is an effective, relatively safe treatment option for patients with CTS who have failed to respond to conservative treatments.

Ultrasonography helps doctors identify the FR and the structures at risk, e.g., the MN, the recurrent motor branch of MN, third common palmar digital nerve and any unusual distal branches, the superficial palmar arch, and the ulnar artery. For safe and effective transection of FR, a transverse safe zone between the hook of the hamate or ulnar vessels and the MN and another longitudinal safe zone between the superficial palmar arch and the distal FR should be identified by ultrasound. The FR, nerves, blood vessels, other at-risk structures, and safe zones could be identified using ultrasound in all the included studies. To enhance the safety of the transection, in addition to ultrasound visualization of the structures at risk of being damaged by the transection, e.g., the palmar cutaneous branch of the MN and the Berrettini communication between the third and fourth common palmar digital nerves, many studies adopted other measures to enlarge the safe zone. Five clinical trials utilized the ZX-One MicroKnife (Sonex Health, Eagen, MN, USA), which has inflatable balloon buffers [80,83,85,86,88]. Three other studies employed K-wires [63,64,81], two with a Penfill curved elevator [85,88], one with a uterine dilator [86], one with a U-shaped trough/probe [66], and one with a button tip cannula with hydrodissection to enlarge the safe zone [89].

#### 4.3.1. Methods of Ultrasound-Guided Percutaneous Carpal Tunnel Release

The approaches for UPCTR also differ in terms of the direction of entry of the instruments and transection devices and the number of passes required for complete transection. Nineteen studies utilized a proximal to distal instrument entry/incision site and three used a distal entry site in the palm proximal to the superficial palmar arch, usually just distal to the distal end of FR [17,71,78]. Thirteen studies used a hook knife as the transection instrument, which is a retrograde blade; in addition, three, three, and three used a needle, a looped thread, and anterograde blades, respectively. Two main approaches have been employed in the studies using a hook knife. In 10 studies from eight groups, the hook knife was positioned deep at the FR and transected the FR with the blade pointing upward [63,64,79,80,81,83,85,86,88,89]. Three clinical studies by the same group placed the hook knife superficial to the FR, with the blade directed downward [76,82,87]. Regarding passes using a hooked knife, one pass of the blade is generally adequate for the complete transection of the FR (85% or more) [85,88]. Kamel et al. [86] reported that nearly 39% of patients needed two passes for a complete transection in the presence of a markedly thickened FR. Among the studies using anterograde blades, Fuente et al. and Nakamichi et al. employed a single passage to divide the FR [17,66], whereas Hebbard et al. [84] used two to three passes to completely transect the FR. The studies using multiple fenestrations with a needle reported requiring 10–15 fenestrations for a complete UPCTR [65]. The studies using a looped thread typically employed a forward and backward sawing motion to transect the FR [71,77,78]. Only one study by Lee et al. [90] used an 18 G needle with the tip bent to release the FR under ultrasound guidance and by repeated cutting. However, no study has directly compared any two different approaches of UPCTR or any two different devices in the literature. Further studies are needed to evaluate their comparative efficacy.

#### 4.3.2. Potential Cost Benefits of UPCTR

One of the potentially significant benefits of UPCTR is the shortened procedural duration, which may lead to a shorter duration of time in the operation theater or the use of alternate settings, both potentially cost-saving. Only five of the 24 included studies described the average duration of the procedure, which ranged from 5.8 to 16.8 min [77,79,84,86,88]. Eleven studies described the procedural setting, with four performed in the operation theater [76,80,85,88], six in an ambulatory clinic procedure room [63,64,80,81,83,132], and one in an interventional radiology procedure room [79]. The timing of return to work was assessed in five studies. Two reported a return to work time of a combined mean of 4 to 5 days in the UPCTR groups compared with 26 days in the mini-OCTR groups [63,64]. Hebbard et al. [84] reported a mean return to work time of 7 days [84]. Guo et al. [78] reported a mean of 17.7 days in their first case series, but in their second case series, the time to return to work was two weeks for manual workers and one day for office workers. Asserson et al. [72] found that the average time of return to work was 12 days in the UPCTR group and 33 days in the OCTR group. Joseph and Leiby et al. suggested that patients with occupations requiring repetitive or heavy use of hands could return to light duties one week after UPCTR [85,88]. Henning et al. [80] reported that three patients using a crutch or wheelchair could ambulate immediately after UPCTR. Chappell et al. [83] advised patients to avoid strenuous activities for four days, whereas Guo et al. [70] immobilized the treated wrists for three days. Asserson et al. [72] showed that participants in the UPCTR group had an average of 12 days to return to work without restriction, whereas those in the OCTR group had an average of 33 days to return to work. A multicenter case series of UPCTR by Fowler et al. [91], including 373 patients (427 hands), reported a rapid median time to return to normal activities (3 days) or work (5 days), in addition to clinically meaningful improvements in symptoms and function.

#### 4.3.3. Other Observations across All Studies Pertaining to Primary Measures and Other Measures Not Utilized for the Meta-Analyses

##### Boston Carpal Tunnel Questionnaire (BCTQ)

Seventeen of the included studies used the BCTQ [133] for pre- and post-procedural assessment of the severity of the symptoms [65,66,71,76,77,78,79,80,81,82,83,84,85,86,87,88,89]. Significant improvements in the BCTQ symptom severity scale and functional status scale were reported as early as one week after UPCTR [78,85,88], and the statistical significance was maintained for up to 2 years [82,87]. The remaining studies showed statistically significantly improved scores from the BCTQ compared with pre-procedural scores. Although significant improvements occurred, as measured by the BCTQ in these studies, the clinical magnitude of improvement may be limited, in that only six of the 17 studies [66,80,83,85,86,88] reported improvements exceeding the reported minimal clinically important difference of 1.14 points for BCTQ-SS and 0.74 points for BCTQ-FS [31,32].

##### Sensory Examination and Grip or Pinch Strength

Among the eight included studies that used sensory examination changes via two-point discrimination or monofilament testing [17,63,64,66,71,76,82,87], seven showed statistically significant improvement from the baseline [17,63,64,66,71,82,87]. The sensory outcomes were statistically similar between the UPCTR and mini-OCTR groups [17,63,64,66]. Seven studies [17,63,64,66,76,82,87] showed long-term statistically significant improvement in hand grip and pinch strength from the pre-procedural state, whereas two studies found no improvement [71,132]. UPCTR outperformed mini-OCTR in terms of hand grip or pinch strength changes for up to 6 weeks post-procedure in two studies [17,64]. However, no long-term differences were detected between the groups in three RCTs [63,64,66].

##### Electrodiagnostic Outcomes

Electrodiagnostic outcomes were evaluated in six studies. Distal motor latencies (DMLs) of MN improved significantly in all six studies [17,65,66,70,71,81] and sensory conduction velocities (SCV) improved significantly in the five studies in which it was measured [17,65,66,70,71]. In the two RCTs comparing UPCTR and mini-CTR for electrodiagnostic outcomes, no differences were found between the UPCTR and mini-CTR groups at long-term follow-up for DML or SCV [17,66]. Guo et al. [70] evaluated UPCTR with or without additional post-procedural corticosteroid injection and showed significantly more DML and SCV improvement in the group receiving post-procedural corticosteroid injection.

##### CSA of the MN

Seven studies employed interval CSA ultrasound measurement of the MN. Pre-treatment MN CSA ranged from 13 to 19 mm^2^ and post-procedure MN CSA ranged from 10 to 15 mm^2^ [65,70,71,81,83,132] across the studies. A statistically significant decrease in MN CSA was reported in five of the seven studies [65,70,81,83,132], in one study CSA changes did not reach statistical significance [71], and one study did not report the statistics of MN CSA changes [83]. Two of them showed an increased diameter of MN at the carpal tunnel exit in the post-procedure follow-up [81,132].

##### Quick Disabilities of the Arm, Shoulder and Hand (Q-DASH)

Six studies [63,64,80,85,86,88] adopted Q-DASH [134] for outcome assessment and five of them reported statistically significant improvement in Q-DASH scores from 1 week [63,64,85] up to 1 year [86,88]. Four of these six studies reported exceeding the MCID for Q-DASH [80,85,86,88] of 15 points post-operatively [135].

#### 4.3.4. Safety and Complications

No significant safety issue and complications were reported in UPIT. Lam et al. has stressed how to prevent damaging the MN during ultrasound-guided nerve hydrodissection [95,99,136,137]. A total of 33 complications were reported among the 2547 wrists that received UPCTR: 13 out of 213 with the loop thread approach, 10 out of 1400 with the hook knife (retrograde blade) method, and 10 out of 148 with anterograde blade use. Kamel et al. [86] defined major complications of UPCTR as nerve, tendon, or vessel injury that required operative management, and by that definition, no major complications were reported. One patient required surgery for acute compartment syndrome post-UPCTR using a hook knife release [86]. However, he reportedly played racquetball on day 10 post-procedure. Of the five patients that required revision surgeries after UPCTR, two patients had persistent symptoms after dovetail blade use, two had persistent symptoms related to incomplete FR transection using the loop thread approach, and one developed recurrent symptoms two years following hook knife UPCTR [66,71,76]. Other mild complications included self-limited pain or swelling in nine patients following looped thread and in two patients following blade UPCTR [77,78,84]. One patient complained of persistent moderate wrist pain without sensory symptoms at one-year post-UPCTR with the hook knife approach [76]. Six patients complained of transient paresthesia after UPCTR with a hook knife, resolving in 1–6 weeks [81,87]. Infections were observed in four patients after UPCTR. Infection occurred in two patients following the inclusion of a corticosteroid in the hydrodissection fluid for looped thread UPCTR [78], in one patient who suffered a fall eight days post-hook knife UPCTR [86], and in another patient following UPCTR with a blade [66].

### 4.4. Limitations and Future Perspectives 

The current study has some limitations. First, all six studies of UPIT with D5W were from a single country, and four were from the same research group, which limited the generalization of results. Second, only a few studies had a follow-up period of more than six months which risks underreporting long-term complications or recurrence and limits the evaluation of the comparative regenerative effects post-injection or surgery. Third, the varying injected volume among the studies may have affected clinical outcomes as a larger volume would be expected to provide greater mechanical hydrodissection. In addition to the shortcomings mentioned above, the current research did not consider all aspects of UPIT and UPCTR. Future studies should clarify other questions regarding UPIT and UPCTR, such as the optimal dosage and frequency of UPIT with different injectates, direct comparisons of effects and efficacy of different devices in UPCTR, and their effect for subgroup patients who have a higher risk of developing CTS, e.g., those with uremia, diabetes mellitus, or rheumatoid arthritis.

## 5. Conclusions

This is the first systematic meta-analysis pertaining to both UPIT and UPCTR. Despite a broad spectrum of bias risks and heterogeneities across the included studies, these results suggest that UPIT with D5W or PRP outperforms the corresponding controls for the treatment of CTS and that UPCTR is at least as effective as open surgery. Both UPIT and UPCTR appear to have safety advantages compared to open surgery. Further studies are needed to determine the relative cost efficacy of these treatment approaches.

## Figures and Tables

**Figure 1 diagnostics-13-01138-f001:**
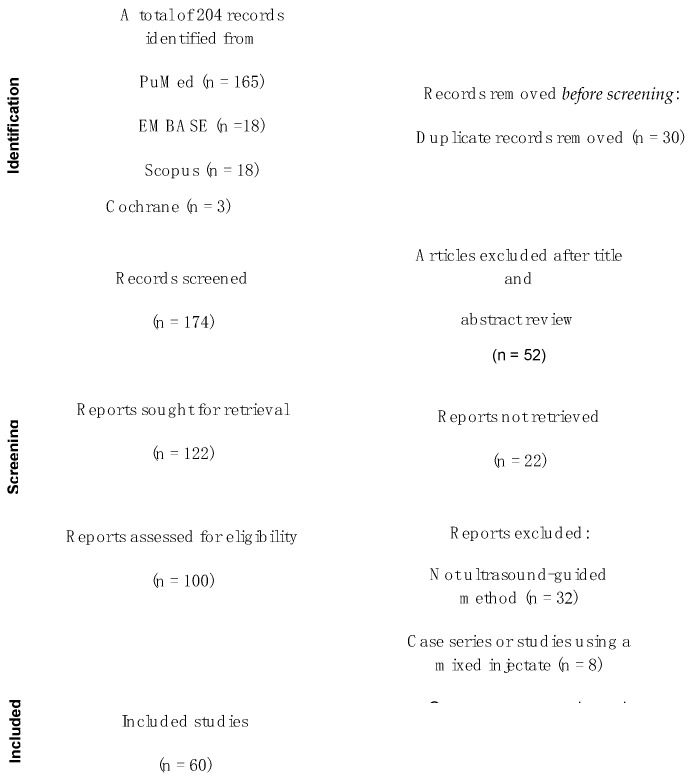
PRISMA flow diagram showing the selection of studies.

**Table 2 diagnostics-13-01138-t002:** JBI Critical Appraisal Checklist for Cohort Studies.

	Criteria and Corresponding Scores													
Author	#1	#2	#3	#4	#5	#6	#7	#8	#9	#10	#11	Total	%	Bias Risk
Bias table for case cohort of UPIT using Steroids														
Hsu (2018) [67]	1	1	1	1	1	1	1	0	0	0	1	8	72.7	Low
Yeom (2021) [68]	0	1	1	0	0	1	1	1	0	0	1	6	54.5	High
Bias table for case cohort of UPIT using Platelet-Rich Plasma														
Guven et al. (2019) [69]	1	1	1	1	0	1	1	0	1	N	1	8	80	Low
Bias table for case cohort of UPCTR														
Nakamichi et al. (2010) [17]	1	1	1	U	U	1	1	1	1	U	1	8	72.7	Low
Guo et al. (2018) [70]	1	1	1	1	1	1	1	0	U	U	1	8	72.7	Low
Burnham et al. (2021) [71]	1	1	1	1	1	1	1	1	1	1	1	11	100	Low
Asserson 2022 [72]	1	1	1	1	1	1	1	1	1	U	1	10	90.0	Low
Quality measures of cohort studies based on the following listed criteria:														
#1. Were the two groups similar and recruited from the same population?														
#2. Were the exposures measured similarly to assign people to both exposed and unexposed groups?														
#3. Was the exposure measured in a valid and reliable way?														
#4. Were confounding factors identified?														
#5. Were strategies to deal with confounding factors stated?														
#6. Were the groups/participants free of the outcome at the start of the study (or at the moment of exposure)?														
#7. Were the outcomes measured in a valid and reliable way?														
#8. Was the follow-up time reported and sufficient to be long enough for outcomes to occur?														
#9. Was follow-up completed, and if not, were the reasons for the loss of follow-up described and explored?														
#10. Were strategies to address incomplete follow-up utilized?														
#11. Was appropriate statistical analysis used?														

NB: 1 indicates the article does fulfill the specified criteria; 0 indicates the article does not fulfill the stated criteria; U indicates the article is unclear about the criteria; N indicates the criteria are not applicable to the article. Red color front signifies high bias, and green color words denote low bias.

**Table 4 diagnostics-13-01138-t004:** Intervention details of included studies using ultrasound-guided corticosteroid injections for treating carpal tunnel syndrome.

Author, Year	Study Design	Inclusion Criteria	UG Intervention and Treatment Allocation	Participant Characteristics				CTS Severity	Outcome Measurements	Follow-Up	Safety Outcome (n)
				Sample Size (Wrists) Cases/Controls	Mean Age (Years) Cases/Controls	Female (%) Cases/Controls	Symptom Duration (Months) Cases/Controls				
Üstün (2013) [33]	Single-blind RCT	Clinical + EDS	UG (Ulnar S–O below MN) vs. Blind (Ulnar to PL) 40 mg methylprednisolone	23/23	45.96/42.71	82.6/95.7	16.78/10.19	Moderate	BCTQ	3 months	Procedural pain UG group (4) Blind group (8)
Lee et al. (2014) [34]	Single–blind RCT	Clinical + EDS	UG (Ulnar S–I below and above MN)) vs. UG (Ulnar S–O below MN) vs. Blind (Ulnar to PL) 1 mL triamcinolone (40 mg/mL) + 1 mL 1% lidocaine	24/26/25	52.6/55.2/50.3	100/93.3/86.7	9.4/8.9/7.6	Mild to moderate	BCTQ EDS CSA of MN	3 months	Nerve insult Ulnar S–I (1) Ulnar S–O (6) Blind (5) Vessel insult Ulnar S–I (0) Ulnar S–O (0) Blind (2) Skin lesion Ulnar S–I (3) Ulnar S–O (1) Blind (8)
Makhlouf et al. (2014) [35]	Single-blind RCT	Clinical	UG (Ulnar S–I above and below MN) vs. Blind (Ulnar to PL) 3 mL 80 mg triamcinolone + 3 mL 1 % lidocaine	37/40	45.7/52.2	94.6/80	NR	Mild to moderate	VAS Duration of therapeutic effect time to next procedure, procedural costs	6 months	No AE reported
Eslamian et al. (2017) [36]	Single-blind RCT	Clinical + EDS	UG (Ulnar S–I below MN) vs. Blind (ulnar to PL) 40 mg methylprednisolone	30/30	54.52/49.33	86.2/100	NR	Moderate	BCTQ EDS,	3 months	1 wrist flexor tendonitis in blind group
Karaahmet al. (2017) [37]	Single-blind RCT	Clinical + EDS	UG (Ulnar S–I, NR about above or below MN) vs. Blind (Ulnar to PL) 1 mL betamethasone (2.63 mg)/betamethasone (6.43 mg)	21/19	59.4/61.5	86.7/93.8	28.5/38.5	Severe	BCTQ EDS	4 weeks	No AE reported
Wang et al. (2017) [38]	Single-blind RCT	Clinical + EDS	Splint + UG (Ulnar S–I above and below MN) vs. UG (Ulnar S–I above and below MN) 1 mL 10 mg (10 mg/mL) triamcinolone + 1 mL 2% lidocaine	24/24	54.34/55.76	83.3/87.5	6/5 (3 to 6 months) 6/7 (6 to 12 months) 6/5 (1 to 2 years) 8/9 (>2 years)	NR	VAS BCTQ EDS Global assessment of treatment results	12 weeks	No AE reported
Chen et al. (2018) [39]	Double-blind RCT	Clinical + EDS	UG (Ulnar S–O below MN) vs. Blind (Ulnar to PL) 1 mL betamethasone (betamethasone dipropionate 5 mg and betamethasone disodium phosphate 2 mg)	22/17	51.09/51.12	76.5/71.4	70.55/65.12	Mild to moderate	Physical findings BCTQ EDS	6 months	UG group Numbness (1) Swelling (4) Pain (10) Weakness (0) Blind group Numbness (4) Swelling (6) Pain (10) Weakness (3)
Babaei-Ghazani et al. (2018) [40]	Double-blind RCT	Clinical + EDS	UG (Ulnar S–I above MN) vs. UG (Ulnar S–I below MN) 1 mL triamcinolone (40 mg/mL)	22/22	56.63/51.09	91/91	NR	Mild to moderate	VAS BCTQ EDS CSA of MN	3 months	No AE reported
Roghani et al. (2018) [41]	Triple-blind RCT	Clinical + EDS	UG (Long-axis from distal to proximal, above MN) 80 mg triamcinolone (2 mL) + 1 mL 2% lidocaine vs. 40 mg triamcinolone (1 mL) + 1 mL 2% lidocaine + 1 mL NS vs. 1 mL 2% lidocaine + 2 mL NS	32/32/30	66.1/66/63.4	68.8/87.5/90	NR	Moderate	VAS BCTQ EDS CSA of MN	6 months	No AE reported
Hsu et al. (2018) [67]	Case cohort	Clinical + EDS + US	UG (Ulnar S–I above MN) Intraepineurium vs. Extraepineurium 3 mL 10 mg triamcinolone + 3 mL 1% lidocaine	39/62	53/56	74.4/75.8	33.2/33.5	NR	BCTQ CSA of MN Satisfaction scale of treatment results	6 months	MN injury Intraepineurium (2) Extraepineurium (4)
Roh et al. (2019) [42]	Single-blind RCT	Clinical + EDS	UG (Ulnar S–I above MN) vs. Blind (Ulnar to PL) 1 mL triamcinolone (20 mg/mL) + 1 mL lidocaine (10 mg/mL)	51/51	54/55	76.5/80.4	15/14	Mild to extremely severe CTS	BCTQ grip strength	6 months	UG group Finger numbness or weakness (1) Skin discoloration or subcutaneous fat atrophy (1) Steroid flare (2) Blind group Finger numbness or weakness (7) Skin discoloration or subcutaneous fat atrophy (3) Steroid flare (3)
Rayegani et al. (2019) [43]	Single-blind RCT	Clinical + EDS	UG (Ulnar S–I below MN) vs. UG (Long-axis from proximal to distal, above MN) vs. Blind 1 mL triamcinolone (40 mg) + 1 mL 2% lidocaine	26/27/23	54.39/54.56/54.04	73.1/81.5/78.3	NR	Mild to moderate	VAS BCTQ grip strength CSA of MN EDS	10 weeks	No AE reported
Hsu et al. (2020) [44]	Double-blind RCT	Clinical + EDS	UG (Ulnar S–I, NR about above or below MN) 40 mg triamcinolone (40 mg/mL) + 1 mL 2% lidocaine vs. 10 mg triamcinolone (10 mg/mL) +1 mL 2% lidocaine	28/28	57.1/54.5	75/78.6	42.3/27.5	Mild to moderate	VAS BCTQ EDS	3 months	No AE reported
Babaei-Ghazani et al. (2020) [45]	Double-blind RCT	Clinical + EDS	UG (Ulnar S–I, below MN) vs. UG (Radial S–I, below MN) 1 cc triamcinolone (40 mg/mL)	30/30	51.7/52.67	NR	NR	Mild to moderate	VAS BCTQ EDS CSA of MN	3 months	No AE reported
Mezian et al. (2021) [46]	Double-blind RCT	Clinical + EDS + US	UG (Ulnar S–I, below MN) Perineural vs. Peritendinous 1 mL methylprednisolone (40 mg/mL) + 1 mL trimecaine hydrochloride	23/23	50/54.3	78.3/82.6	5.9/5.9	At least mild CTS	VAS BCTQ, Physical findings Grip strength EDS CSA of MN	3 months	No AE reported
Yeom et al. (2021) [68]	Retrospective study	Clinical + EDS + US	UG (Radial S–I, below MN) 1 mL triamcinolone (40 mg/mL) + 1 mL of 1% lidocaine	40	59.6	77.5	15.8	NR	BCTQ Q-DASH, percentage of treatment failure	Mean 16 months (range 7 to 43 months)	No AE reported
Wang (2021) [47]	Single-blind RCT	Clinical + EDS	UG (Ulnar S–I, above and below MN) 1 mL triamcinolone (10 mg/mL) + 1 mL 2% lidocaine + 8 mL NS vs. 1 mL triamcinolone (10 mg/mL) + 1 mL 2% lidocaine	32/32	52.87/53.28	75/87.5	22.93/24.31	Mild to moderate	BCTQ EDS	12 weeks	2 patients in the hydrodissection group reported minor post-injection pain on the first day after the intervention that resolved spontaneously
Mathew et al. (2022) [48]	Open-label parallel RCT (non-blind)	Clinical + EDS	UG (Ulnar S–I, around MN) Dexamethasone 8 mg (2 mL) + 2 mL 0.5% bupivacaine) vs. Triamcinolone 40 mg/mL (1 mL) + 2 mL 0.5% bupivacaine + 1 mL NS	33/36	42.64/45.22	80.6/87.1	NR	Mild to moderate	VAS BCTQ EDS Phalen’s test	4 months	No AE reported

CTS: carpal tunnel syndrome; RCT: randomized controlled trial; UG: ultrasound-guided; NS: normal saline; VAS: visual analog scale; NR: not reported; AE: adverse effect; BCTQ: Boston Carpal Tunnel Syndrome Questionnaire; EDS: electrodiagnostic study; CSA: cross-sectional area; MN: median nerve; 2PD: two-point discrimination; Ulnar S–O: ulnar short-axis out-of-plane; Ulnar S–I: ulnar short-axis in-plane; Ulnar S–O: ulnar short-axis out-of-plane; Radial S–I: radial short-axis in-plane; PL: palmaris longus tendon; US: ultrasound; DASH: Disabilities of the Arm, Shoulder and Hand.

**Table 5 diagnostics-13-01138-t005:** Intervention details of included studies using ultrasound-guided D5W injection for treating carpal tunnel syndrome.

Author, Year	Study Design	Inclusion Criteria	UG Intervention and Treatment Allocation	Participant Characteristics				CTS Severity	Outcome Measurements	Follow-Up	Safety Outcome (n)
				Sample Size (Wrists) Cases/Controls	Mean Age (Years) Cases/Controls	Female (%) Cases/Controls	Symptom Duration (Months) Cases/Controls				
Wu et al. (2017) [49]	Double-blind RCT	Clinical + EDS	UG (Ulnar S–I below and above MN) 5 mL D5W vs. 5 mL NS	30/30	58.4/58.1	86.7/80	44.5/44.4	Mild to moderate	VAS BCTQ EDS CSA of MN Global assessment of treatment results	6 months	No AE reported
Wu et al. (2018) [50]	Double-blind RCT	Clinical + EDS	UG (Ulnar S–I below and above MN) 5 mL D5W vs. 3 mL triamcinolone (10 mg/mL) + 2 mL NS	27/27	58.6/54.3	81.4/77.7	46.8/45.6	Mild to moderate	VAS BCTQ EDS CSA of MN Global assessment of treatment results	6 months	No AE reported
Lin et al. (2020) [51]	Randomized, double-blind, three-arm trial	Clinical + EDS	UG (Radial S–I below and above MN) 4 mL D5W vs. 2 mL D5W vs. 1 mL D5W	21/21/21	58.4/55.2/60.3	95.2/81/81	54.4/20.6/49.8	NR	VAS BCTQ Q-DASH EDS CSA of MN	6 months	No AE reported
Lin et al. (2021) [52]	Randomized, double-blind, three-arm trial	Clinical + EDS	UG (Radial S–I below and above MN) 4 mL D5W vs. 2 mL D5W vs. 1 mL D5W	17/14/14	56.9/52.9/59.2	94.1/85.7/85.7	66/21.9/58.4	NR	Mobility, shear wave elastography CSA of MN VAS BCTQ	6 months	NR
Li et al. (2021) [73]	Retrospective study	Clinical + EDS	Mean 2.2 UG injections with 10 mL D5W (Ulnar S–I below and above MN + L–I from proximal to distal)	185	55.4	65.4	30.8	All grades	VAS Surgical rate	At least 1 year (1–3 years) post-injection (mean 15.8 months)	No AE reported
Chao et al. (2022) [74]	Retrospective study	Clinical + EDS	Mean 3.1 UG injections with 10 mL D5W (Ulnar S–I below and above MN + L–I from proximal to distal)	36	59.2	77.8	15.1	Persistent or recurrent CTS after surgery	VAS	At least 6 months (6–67 months) post-injection (mean 33 months)	No AE reported

CTS: carpal tunnel syndrome; RCT: randomized controlled trial; VAS: visual analog scale; NR: not reported; AE: adverse effect; BCTQ: Boston Carpal Tunnel Syndrome Questionnaire; NS: normal saline; EDS: electrodiagnostic study; CSA: cross-sectional area; MN: median nerve; Q-DASH: Quick Disabilities of the Arm, Shoulder and Hand score; Ulnar S–I: ulnar short-axis in-plane; Radial S–I: radial short-axis in-plane; D5W: 5% dextrose in water.

**Table 6 diagnostics-13-01138-t006:** Intervention details of included studies using ultrasound-guided PRP injection for treating carpal tunnel syndrome.

Author, Year	Study Design	Inclusion Criteria	UG intervention and Treatment Allocation	Participant Characteristics				CTS Severity	Outcome Measurements	Follow-Up	Safety Outcome (n)
				Sample Size (Wrists) Cases/Controls	Mean Age (Years) Cases/Controls	Female (%) Cases/Controls	Symptom Duration (Months) Cases/Controls				
Malahias et al. (2015) [75]	Pilot study (Case series)	NR	UG (Ulnar S–I below MN) 1–2 mL PRP	14/0	61.5	92%	NR (Minimum of 3-month duration of symptoms)	Mild to moderate	Q-DASH VAS	3 months	No AE reported
Wu et al. (2017) [53]	Single-blind RCT	Clinical + EDS	UG (Ulnar S–I below and above MN) 3 mL PRP vs. Splint	30/30	57.87/54.27	90%/8 3.3%	34.43/30.7	Mild to moderate	VAS BCTQ EDS CSA of MN Finger pinch strength	6 months	No AE reported
Malahias et al. (2018) [54]	Double-blind RCT	Clinical	UG (Ulnar S–I below MN) 2 mL PRP vs. NS	26/24	60.4/57.1	NR	NR (Minimum of 3-month duration of symptoms)	Mild to moderate	VAS Q-DASH Delta- CSA of MN	3 months	No AE reported
Guven et al. (2019) [69]	Prospective quasi-experimental	Clinical + EDS	UG (above MN, no mentioned approach side) 1 mL PRP + splint vs. Splint	20/20	47.5/50	94.4/91.6	72/60	Mild to moderate	BCTQ EDS CSA of MN Monofilament testing score Static 2PD testing score Dynamic 2PD testing score	4 weeks	No AE reported
Senna et al. (2019) [55]	Single-blind RCT	Clinical + EDS	UG (Ulnar S–I above MN) 2 mL PRP vs. Corticosteroid	43/42	38.3/40.7	81.4/85.7	NR	Mild to moderate	VAS BCTQ EDS CSA of MN Paresthesia Phalen’s maneuver Tinel’s sign	3 months	No AE reported
Shen et al. (2019) [56]	Single-blind RCT	Clinical + EDS	UG (Ulnar S–I below and above MN) 3 mL PRP vs. 3 mL D5W	26/26	56.8/58.5	96.2/84.6	58.3/37.5	Moderate	BCTQ EDS CSA of MN	6 months	No AE reported
Chen et al. (2021) [57]	Double-blind RCT	Clinical + EDS	UG (Ulnar S–I below and above MN) 3.5 mL PRP vs. NS	24/24	53/53	87.5/87.5	35.3/36.2	Moderate to severe	BCTQ EDS CSA of MN	1 year	No AE reported

CTS: carpal tunnel syndrome; RCT: randomized controlled trial; Q-DASH: Quick Disabilities of the Arm, Shoulder and Hand score; VAS: visual analog scale; NR: not reported; AE: adverse effect; BCTQ: Boston Carpal Tunnel Syndrome Questionnaire; VAS: visual analog scale; PRP: platelet-rich plasma; D5W: 5% dextrose in water; EDS: electrodiagnostic study; CSA: cross-sectional area; MN: median nerve; 2PD: two-point discrimination; Delta-CSA: cross-sectional area of the median nerve at the tunnel’s inlet, minus the median nerve, proximal to the tunnel and overpronator quadratus.

**Table 7 diagnostics-13-01138-t007:** Intervention details of included studies using ultrasound-guided hyaluronic acid, hyaluronidase, insulin, and ozone injection for treating carpal tunnel syndrome.

Author, Year	Study Design	Inclusion Criteria	UG Intervention and Treatment Allocation	Participant Characteristics				CTS Severity	Outcome Measurements	Follow-Up	Safety Outcome (n)
				Sample Size (Wrists) Cases/Controls	Mean Age (Years) Cases/Controls	Female (%) Cases/Controls	Symptom Duration (Months) Cases/Controls				
Su et al. (2021) [58]	Double-blind RCT	Clinical + EDS	UG (Long-axis from proximal to distal, above MN) 2.5 mL HA vs. 2.5 mL NS	17/15	50.9/58.9	76.5/80	35.6/28.6	Mild to moderate	NRS BCTQ EDS CSA of MN	6 months	No AE reported
Alsaeid et al. (2019) [59]	Double-blind RCT	Clinical + EDS + US	UG (Ulnar S–I above and below MN) 300 units IU hyaluronidase in 2 mL NS + 3 mL 0.5% plain bupivacaine vs. 2 mL (8 mg) dexamethasone) + 3 mL 0.5% plain bupivacaine	20/20	40.18/42.76	55/50	NR	Mild to moderate	BCTQ EDS CSA of MN Echogenicity score + mobility score + vascularity score of MN	6 months	NR
Elawamy et al. (2020) [60]	Double-blind RCT	Clinical + EDS	UG (Ulnar S–I above MN) 1500 IU hyalase in 10 mL NS vs. 10 mL NS	30/30	40.7/38.3	56.7/56.7	8.5/8.5	Mild to moderate	VAS Modified BCTQ EDS CSA Power Doppler of MN	6 months	No AE reported
Kamel et al. (2019) [61]	Single-blind RCT	Clinical + EDS	UG (Ulnar S–I above MN) 10 IU insulin * 2 times (2 weeks interval) vs. 40 mg methylprednisolone vs. 40 mg methylprednisolone + 10 IU insulin * 2 times after 2 and 4 weeks	20/20/20	40.7/44.7/38.3	85/90/90	8.5/8.1/7.5	Mild to moderate	Modified BCTQ EDS Physical finding CSA Power Doppler of MN	10 weeks	NR
Forogh et al. (2021) [62]	Double-blind RCT	Clinical + EDS	UG (Ulnar S–I above MN) 3 mL ozone (O2–O3) (10 μg/mL) + 1 mL lidocaine vs. 40 mg triamcinolone + 1 mL lidocaine	20/20	54.7/53.65	NR	9.1/10.85	Mild to moderate	VAS BCTQ EDS Circumference and CSA of MN	3 months	NR

CTS: carpal tunnel syndrome; RCT: randomized controlled trial; NRS: numeric rating scale; VAS: visual analog scale; NR: not reported; US: ultrasound; AE: adverse effect; BCTQ: Boston Carpal Tunnel Syndrome Questionnaire; NS: normal saline; UG: ultrasound-guided; EDS: electrodiagnostic study; CSA: cross-sectional area; MN: median nerve; Ulnar S–I: ulnar short-axis in-plane; HA: hyaluronic acid.

**Table 8 diagnostics-13-01138-t008:** Summary of the intervention details in the included studies using ultrasound-guided percutaneous carpal tunnel release.

Author, Year	Study Design	Inclusion Criteria	UG Intervention (Device) and Treatment Allocation	Participant Characteristics				CTS Severity	Outcome Measurements	Follow-Up	Safety Outcome
				Sample Size (Wrists) Cases/Controls	Mean Age (Years) Cases/Controls	Female (%) Cases/Controls	Symptom Duration (Months) Cases/Controls				
Nakamichi et al. (2010) [17]	Controlled trial	Clinical + EDS	UPCTR (NR device) vs. Mini-OCTR	25/39	58 (all patients)	100 (all patients)	NR	NR	EDS Sensibility (static 2-point discrimination, monofilament) Grip and key pinch strength Pain Scar sensitivity	24 months	No AE reported
Capa-Grasa et al. (2014) [63]	RCT	Clinical + EDS	UPCTR (Acufex 3.0 mm hook knife) vs. Mini-OCTR	20/20	63/58	90/85	37/38	NR	Q-DASH Grip strength, time to stopping oral analgesics, complete wrist flexion and extension, relieving paresthesia, and returning to normal daily activities	3 months	No AE reported
Chern et al. (2015) [76]	Case series	Clinical	UPCTR (custom-made hook knife)	91	58	77.5	48	NR	BCTQ Sensibility (2-point discrimination, monofilament) Grip, key pinch, and three-jaw chuck pinch strength	12 months	No AE reported
Guo et al. (2015) [77]	Case series	Clinical	UPCTR (GuoPercutaneousWire™ looped thread)	34	52	60	>12 months	NR	BCTQ	3 months	Self-limited wrist swelling 3 weeks after the procedure (1)
Rojo-Manaute et al. (2016) [64]	RCT	Clinical + EDS	UPCTR (Acufex 3.0 mm hook knife) vs. Mini-OCTR	46/46	58/59	58.7/63	36/36	NR	Q-DASH Grip strength, pain scores, time to stopping oral analgesics, complete wrist flexion and extension, 2-point discrimination, relieving paresthesia, and returning to normal activities	12 months	UPCTR group No AE reported Mini-OCTR Group CRPS (2) Superficial infection (1)
Guo et al. (2017) [78]	Case series	Clinical + EDS + US	UPCTR (loop and shear looped thread)	159	54.83	66.3	Most >1 year	NR (either failed conservative treatment or requested a surgical release)	BCTQ	12 months	Infection (2) Self-limited pillar pain at 2–6 weeks (8)
Petrover et al. (2017) [79]	Prospective, open-label study	Clinical + EDS	UPCTR (Acufex 3.0 mm hook knife)	129	61.5	69.7	>6 months	NR (failed on conservative treatment)	BCTQ	6 months	No AE reported
Guo et al. (2018) [70]	Controlled trial	Clinical + EDS + US	UPCTR (22 G hypodermic needle) vs. UPCTR + CSI	25/25	50.52/48.64	79.1/68	20.92/19.32	Early to middle stage	Global assessment of treatment results CSA of MN EDS	3 months	No AE reported
Henning et al. (2018) [80]	Case series	Clinical + EDS	UPCTR (SX-One MicroKnife)	22	64	NR	NR	NR	BCTQ Q-DASH	3 months	No AE reported
Luanchumroen (2019) [81]	Case series	Clinical + EDS	UPCTR (Acufex 3.0 mm hook knife)	20	55	87.5	>6 months	Moderate to severe	BCTQ CSA of MN EDS	6 months	Transient paresthesia for 1–2 weeks (5)
Wang et al. (2019) [82]	Case series	Clinical + EDS	UPCTR (hook knife)	113	61	66.6	24	Hemodialysis patient who had failed on conservative treatment >3 months	BCTQ Sensibility (2-point discrimination, monofilament) Grip and pinch strength	2 years	No AE reported
Zhang et al. (2019) [65]	RCT	Clinical + EDS	UPCTR (Hanzhang miniscalpel needle) + CSI vs. CSI	23/23	48.7/53.1	78.2/73.9	10.2/11.1	NR	BCTQ EDS CSA of MN	12 weeks	No AE reported
Chappell et al. (2020) [83]	Case series	Clinical	UPCTR (SX-One MicroKnife)	37	62	30.4	26% <1 year 39% 1 to 5 years 31% >5 years 4% NR	Severe or failed on conservative treatment	BCTQ CSA of MN	10 weeks	No AE reported
Hebbard et al. (2020) [84]	Case series	Clinical + EDS	UPCTR (MICROi-Blade)	166	57	46.3	NR	NR	BCTQ Days of return to work	6 months	Self-limited post-operative numbness for 1–3 weeks (several) Post-operative swelling resolved with CSI (2)
Joseph et al. (2020) [85]	Case series	Clinical + US	UPCTR (SX-One MicroKnife)	35	60	59.1	NR	NR (Failed on conservative treatment >6 months)	BCTQ Q-DASH	3 months	No AE reported
Burnham et al. 2021 [71]	Controlled trial	Clinical + EDS + US	UPCTR (sterile uncoated multifilament stainless steel wire looped thread) vs. No intervention	40/20	NR	55	>3 months	Moderate to severe	BCTQ CSA of MN EDS Sensibility (Semmes–Weinstein monofilaments) Grip and pinch strength)	6 months	No AE reported
Kamel et al. (2021) [86]	Case series	Clinical + US	UPCTR (SX-One MicroKnife)	61	61	54.3	NR	NR (Failed on conservative treatment >6 months)	BCTQ Q-DASH	>1 year (median: 20 months)	Infection after a fall on open wound on post-op day 8 (1) Post-traumatic compartment syndrome on post-op day 10 after wrist injury (1)
Wang et al. (2021) [87]	Case series	Clinical + EDS	UPCTR (ECTRA or E-Z knife hook knife)	641	60	64.1	29	NR (Failed on conservative treatment >3 months or thenar muscle atrophy or weakness)	BCTQ Sensibility (2-point discrimination) Grip strength	24 months	Transient nerve palsy for 6 week (1)
Leiby et al. (2021) [88]	Case series	Clinical + US	UPCTR (SX-One MicroKnife)	76	58	57.4	NR	NR	BCTQ Q-DASH	12 months	No AE reported
Fuente et al. (2021) [66]	Open RCT	Clinical + EDS	UPCTR (U-shaped probe/ trough + 5 mm dovetail blades) vs. OCTR	47/42	46.7/49.1	51.1/57.1	NR	NR	BCTQ Sensibility (2-point discrimination) Grip strength	12 months	UPCTR group epineural fibrosis (1) OCTR group CRPS (1)
Loizides et al. (2021) [89]	Case series	Clinical + EDS	UPCTR (button tip cannula + hook knife)	104	60.6	64.4	NR	NR	US of FR Simplified BCTQ	2 weeks	Sparse hematoma at the incision site
Lee et al. (2021) [90]	Prospective case series	Clinical	UPCTR (18 G needle tip bent in the opposite direction to the needle bevel)	188	54.7	71	50.3 weeks	NR (Failed on conservative treatment)	NRS	6 months	No AE reported
Fowler (2022) [91]	Multicenter observational study	Clinical + EDS	UPCTR (UltraGuideCTR device)	427	55	71	11.6% ≤6 months 21.6% >6 months to 1 year 15.1% >1 year to 2 years 51.7% >2 years	NR	Q-DASH BCTQ Time to return to normal activities	6 months	Incomplete release (1)
Asserson et al. 2022 [72]	Retrospective study	Clinical + EDS	UPCTR (NR device) vs. OCTR	18/17	52.1/47.3	72.2/88.2	NR	NR	Day of return the work	52 weeks	No AE reported

CTS: carpal tunnel syndrome; RCT: randomized controlled trial; NRS: numeric rating scale; VAS: visual analog scale; NR: not reported; US: ultrasound; AE: adverse effect; BCTQ: Boston Carpal Tunnel Syndrome Questionnaire; NS: normal saline; UG: ultrasound-guided; EDS: electrodiagnostic study; CSA: cross-sectional area; MN: median nerve; Q-DASH: Quick Disabilities of the Arm, Shoulder and Hand score; UPCTR: ultrasound-guided percutaneous carpal tunnel release; FR: flexor retinaculum; OCTR: open carpal tunnel release; CSI: corticosteroid injection.

## Data Availability

Data are available within the manuscript.

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
