# Peer review of "Ultrasound-Guided Interventions for Carpal Tunnel Syndrome: A Systematic Review and Meta-Analyses"

_diagnostics, 2023, doi:10.3390/diagnostics13061138_

Round 1

Reviewer 1 Report

There are a number of grammatical and editorial mistakes throughout the manuscript. Therefore, editing of English language is required for the manuscript to ensure consistency and accuracy in grammar, punctuation, and etc…

Thanks a lot for reviewing this artical

Diagnostics is an international, peer-reviewed, open access journal on medical diagnosis published monthly online by MDPI. Section:Medical Imaging and Theranostics.This artical (Ultrasound-guided intervention for carpal tunnel syndrome)is treatment of carpal tuunel syndrome,this injection is now listed as one of the treatment options by UpToDate[60]. The 20th edition of Harrison's Principles of Internal Medicine also officially lists this method as an alternative treatment for the treatment of CTS[61].

Table need summary and organization

UPIT with D5W with multiple injection sections (mean 2.2 injections with 10 mL D5W),1.7 to 3 injections to achieve an effective outcome, and only 1% (2/185) of the patients ultimately underwent surgery.

PRP is rich in various growth factors which can help neural repair and regeneration, as shown by animal and in vitro studies.but RPR must be used at own expense.

The artical is too long and tired to read,can arouse the reader's interest?must summary

Author Response

Comments to the Author

Thanks a lot for reviewing this article

Diagnostics is an international, peer-reviewed, open access journal on medical diagnosis published monthly online by MDPI. Section:Medical Imaging and Theranostics. This artical (Ultrasound-guided intervention for carpal tunnel syndrome)is treatment of carpal tuunel syndrome, this injection is now listed as one of the treatment options by UpToDate. The 20th edition of Harrison's Principles of Internal Medicine also officially lists this method as an alternative treatment for the treatment of CTS.

Question (1)

There are a number of grammatical and editorial mistakes throughout the manuscript. Therefore, editing of English language is required for the manuscript to ensure consistency and accuracy in grammar, punctuation, and etc…

Response:

Thanks for pointing these out. The manuscript has been edited again for language by Editage, a division of Cactus Communications.

Question (2)

Table need summary and organization

Response:

Thanks for your comment. We have summarized and organized all tables per your comment. Lines 173 to 233.

Question (3)

UPIT with D5W with multiple injection sections (mean 2.2 injections with 10 mL D5W), 1.7 to 3 injections to achieve an effective outcome, and only 1% (2/185) of the patients ultimately underwent surgery.

Response:

Thanks for your comment. We have revised this sentence accordingly. Page 45 lines 518 to 521.

Question (4)

PRP is rich in various growth factors which can help neural repair and regeneration, as shown by animal and in vitro studies but RPR must be used at own expense.

Response:

Thanks for your comment. We agree with it. We have revised the text and clarified our intended meaning accordingly. (Page 46, lines 545 to 546)

Question (5)

The artical is too long and tired to read, can arouse the reader's interest? must summary

Response:

Thanks for your comment. We have revised this manuscript and reduced the word count as you suggested. But since this manuscript has been changed to a systemic review and meta-analysis, some essential information do increase the word count.

Reviewer 2 Report

1.     Please include a graphical abstract

2.     State the research question and objective(s) of the systematic review

3.     Some spelling errors in Figure 1.

4.     Use updated version of Prisma flow diagram: https://prisma-statement.org/prismastatement/flowdiagram.aspx

5.     Indicate the start date of systematic search

6.     Inclusion and exclusion criteria should be clearly stated in the text by numbering them

7.     Link the search terms together using logical Boolean operators.

8.     Please ensure abbreviations used in the Tables must be stated in full form in footnotes. 

9.     Inconsistent font size - smaller font size in Table 5

10.  Tables 1-5 shall be mentioned in the Discussion

11.  Are the following sections part of the Discussion? These have not been numbered

Other assessments of treatment outcomes

Electrodiagnostic outcomes 

12.  To include a section of Limitations and Future Perspectives 

Inclusion 

Author Response

Reviewer: 2

Question (1)

Please include a graphical abstract

Response:

Thanks for your suggestion. We will add a graphical abstract according to the guidelines of the journal, if this manuscript has been accepted.

Question (2)

State the research question and objective(s) of the systematic review

Response:

Thanks for your suggestion. We have added the research question and objectives of the systematic review in the revised version of the manuscript. (Page 3, lines 66-69)

Question (3)

Some spelling errors in Figure 1.

Response:

Thanks for pointing out this. We have revised Figure 1 accordingly.

Question (4)

Use updated version of Prisma flow diagram: https://prismastatement.org/prismastatement/flowdiagram.aspx

Response:

Thanks for your suggestion. We have used the updated version of Prisma flow diagram in the revised version of the manuscript accordingly.

Question (5)

Indicate the start date of systematic search

Response:

Thanks for your suggestion. We have added the start date of the systematic search in the revised version of the manuscript. (Page4, lines77-78)

Question (6)

Inclusion and exclusion criteria should be clearly stated in the text by numbering them

Response:

Thanks for your suggestions. We have added the inclusion and exclusion criteria in the revised version of the manuscript. (Page 4, lines 82-86)

Question (7)

Link the search terms together using logical Boolean operators.

Response:

Thanks for your suggestion. We have linked them accordingly.

Question (8)

Please ensure abbreviations used in the Tables must be stated in full form in footnotes. 

Response:

Thanks for your suggestion. We have ensured that all abbreviations are stated in full form in footnotes in all tables.

Question (9)

Inconsistent font size - smaller font size in Table 5

Response:

Thanks for pointing this out. We have revised the font size for all tables to ensure uniformity.

Question (10)

Tables 1-5 shall be mentioned in the Discussion

Response:

Thanks for your suggestion. We have mentioned Tables 1–5 in the discussion section of the revised version of the manuscript.

Question (11)

Are the following sections part of the Discussion? These have not been numbered

Other assessments of treatment outcomes

Electrodiagnostic outcomes

Response:

Thanks for your suggestions. These are parts of the discussion section. We have numbered these parts accordingly.

Other assessments of treatment outcomes and Electrodiagnostic outcomes page 53, lines 711 to 745.

page 57, lines 722 to 730.

Question (12)

To include a section of Limitations and Future Perspectives.

Response:

Thanks for your suggestions. We have added limitations and future perspectives to the revised version of the manuscript. (Page 54, lines 766 to 778)

Reviewer 3 Report

Although interesting, the paper need to be hardly improved

- the abstract need to be improved: where are methods, results and conclusions?
- pay attention to english grammar, syntax rules and punctuation all over the manuscript
- pay attention to the layout of the manuscript
- at the end of the introduction, you have to write the aim of the paper
- always explain acronyms
- in the method section, specify your PICO (patient - intervention - comparison - outcome)
- there are too many tables and these are graphycally wrong: please reduce it
- there is no statistical section
- there is no risk of bias section
- results are too poor

Author Response

Reviewer 3

Comments to the Author

Although interesting, the paper need to be hardly improved

Question (1)

The abstract need to be improved: where are methods, results and conclusions?
Response:

Thanks for your suggestions. We have revised the abstract accordingly.

Question (2)

Pay attention to English grammar, syntax rules and punctuation all over the manuscript
Response:

Thanks for your comment. The manuscript has been edited again for language by Editage, a division of Cactus Communications.

Question (3)

Pay attention to the layout of the manuscript
Response:

Thanks for your suggestion. We have ensured that all formatting guidelines, as provided in the template, are incorporated in our revised manuscript accordingly.

Question (4)

At the end of the introduction, you have to write the aim of the paper
Response:

Thanks for your suggestion. We have mentioned the aim of the study at the end of the introduction section of the revised version of the manuscript. (Page 3, lines 66 to 69)

Question (5)

Always explain acronyms
Response:

Thanks for your comment. We have made the necessary changes accordingly.

Question (6)

In the method section, specify your PICO (patient - intervention - comparison - outcome)
Response:

Thanks for your suggestions. We have revised the text accordingly.

Question (7)

There are too many tables and these are graphycally wrong: please reduce it
Response:

Thanks for your suggestions. We have revised all tables as you suggested. In order to clearly illustrate the related findings, we maintain 5 tables in the revised version of the manuscript and we have added 3 bias assessment tables.

Question (8)

Tables 1-5 shall be mentioned in the Discussion

Response:

Thanks for your suggestion. We have mentioned Tables 4–8 (after adding the 3 bias tables) in the discussion section in the revised version of the manuscript. We have also mentioned all the 3 bias tables in the discussion as well.

Question (9)

There is no statistical section

Response:

Thanks for pointing out this. We have added the statistical section to the revised version of the manuscript. (Page 41 to 47, line 235 to 414)

Question (10)

There is no risk of bias section
Response:

Thanks for pointing out this. We have added the risk of bias section in the revised version of the manuscript as you suggested. (Page 7 to 14, lines126 – 168)

Question (11)

Results are too poor
Response:

Thanks for the valuable comment. We have thoroughly revised the results section to improve clarity and readability. (Page 5 to 6, lines 114 to 123)

Round 2

Reviewer 1 Report

Ultrasound-guided intervention for carpal tunnel syndrome: A systematic review

must summary and let it easy to read for reader.

It is not suitable for publication

Author Response

Thank you for your review and suggestions, much appreciated. 

We have moved the tables 1 to 8 to the supplementary materials and we have made necessary changes to the tables to make it easier to read. 

Reviewer 3 Report

The article is now improved

Author Response

Thank you for your review, much appreciated. 

We have further improve the readability of the manuscript by moving the tables 1 to 8 to the supplementary materials and made necessary changes to the table 2 and 3.